# Epidemiological Analysis of Diabetes-Related Hospitalization in Poland before and during the COVID-19 Pandemic, 2014–2020

**DOI:** 10.3390/ijerph191610030

**Published:** 2022-08-14

**Authors:** Kuba Sękowski, Justyna Grudziąż-Sękowska, Paweł Goryński, Jarosław Pinkas, Mateusz Jankowski

**Affiliations:** 1School of Public Health, Centre of Postgraduate Medical Education, 01-826 Warsaw, Poland; 2Department of Population Health Monitoring and Analysis, National Institute of Public Health-National Institute of Hygiene, 00-791 Warsaw, Poland

**Keywords:** diabetes mellitus, type 1 diabetes, type 2 diabetes, hospitalization, hospital admission, in-hospital mortality, epidemiology, COVID-19 pandemic, Poland

## Abstract

Diabetes is one of the most common chronic diseases worldwide. The study aimed to present an epidemiological analysis of hospitalization related to diabetes mellitus in Poland between 2014 and 2020 as well as to analyze changes in diabetes-related hospital admissions before and during the COVID-19 pandemic. This study is a retrospective analysis of the national registry dataset of hospital discharge reports on diabetes-related hospitalizations in Poland between 2014 and 2020. The number of diabetes-related hospitalizations varied from 76,220 in 2016 to 45,159 in 2020. The hospitalization rate per 100,000 has decreased from 74.6 in 2019 to 53.0 in 2020 among patients with type 1 diabetes (percentage change: −28.9%). An even greater drop was observed among patients with type 2 diabetes: from 99.4 in 2019 to 61.6 in 2020 (percentage change: −38%). Both among patients with type 1 and type 2 diabetes, a decrease in hospitalization rate was higher among females than males (−31.6% vs. −26.7% and −40.9% vs. −35.2% respectively). When compared to 2019, in 2020, the in-hospital mortality rate increased by 66.7% (60.0% among males and 65.2% among females) among patients hospitalized with type 1 diabetes and by 48.5% (55.2% among females and 42.1% among males) among patients hospitalized with type 2 diabetes. Markable differences in hospitalization rate, duration of hospitalization, as well as in-hospital mortality rate by gender, were observed, which reveal health inequalities.

## 1. Introduction

Diabetes mellitus is a group of chronic metabolic disorders characterized by a high blood sugar level (hyperglycemia), which results from abnormalities in either insulin secretion or action [1,2,3]. The most common forms of diabetes are type 1 diabetes, where complete insulin deficiency leads to autoimmune destruction of the pancreatic islet beta cells (insulin-dependent diabetes), and type 2 diabetes, where cells do not respond normally to insulin (insulin resistance) [1,2,3]. It is estimated that 90–95% of people with diabetes have type 2 diabetes [4,5].

Diabetes is one of the most common chronic diseases worldwide [6]. The World Health Organization (WHO) stated that diabetes is a problem in society and may be called the pandemic of the 21st century [7]. It is estimated that 8.5% of the world’s population has diabetes [7,8]. According to the International Diabetes Federation estimates, the global number of people with diabetes will increase from 425 million in 2017 to 629 million by 2045 [8].

More than three-quarters of adults with type 2 diabetes live in low- and middle-income countries [7,8]. Urbanization, income status, population growth, and aging are the major factors affecting the current growth in the prevalence of diabetes [9,10]. Moreover, lifestyle changes such as a high-energy diet combined with limited physical activity (sedentary lifestyle) significantly increase the risk for type 2 diabetes [11]. Moreover, there are markable regional differences in the incidence of type 1 diabetes [9,10]. The incidence of type 1 diabetes began to increase in the 1950s, with an average annual increase of 3–4% over the past three decades [12,13,14].

It is estimated that over 33 million people (6.3% of the population) in the European Union (EU) suffer from diabetes [15]. Between 2000 and 2019, an almost twofold increase in the number of diabetic patients was observed [15]. Diabetes prevalence among adults in the EU varies from 15.3% in Germany to 4.4% in Ireland [16]. Poland is the fifth most populous EU country, with 38 million citizens of which approximately 8% have diabetes [15,17]. When compared to other EU countries, Poland is a country with a relatively high percentage of patients with diabetes-related complications [18]. Between 2014–2018, the number of limb amputations related to diabetes increased by 20 percentage points [18]. Effective management of diabetes is one of the major challenges for the healthcare system in Poland.

Clinical management of type 2 diabetes is complex and includes lifestyle changes (healthy diet, regular exercise, weight loss), regular blood sugar monitoring, and management of comorbidities (mostly cardiovascular diseases or dyslipidemia) [19]. Moreover, if non-pharmacological interventions are insufficient to maintain target blood sugar levels, diabetes medications (with metformin as a first-choice drug) should be prescribed [19,20]. Clinical management of type 1 diabetes is based on pharmacotherapy (insulin therapy) [21,22].

Most of the international scientific societies’ guidelines on diabetes recommend the treatment of diabetes in an outpatient setting [19,20,21,22]. Hospital-based treatment should be offered only for those patients who have serious diabetes-related complications, mostly due to low compliance with treatment regimens or emergencies [19,21,23]. However, there are markable differences in the diabetes treatment patterns across the EU countries [24]. For example, the guidelines of the Polish Diabetes Association on the management of diabetes are mostly consistent with the international guidelines, with some exemptions. The national guidelines on the management of diabetes recommend that each patient newly diagnosed with type 1 diabetes should be hospitalized in a specialist diabetes unit [25].

Global changes in the incidence of diabetes pose a significant challenge for the national healthcare systems [4,6,8]. The capacity of national healthcare systems should be adjusted to the current health needs of diabetic patients. Moreover, the outbreak of the COVID-19 pandemic had a markable impact on healthcare utilization among patients with diabetes [26]. Anti-epidemic responses led to several restrictions and organizational changes in the healthcare system [27]. For example, scheduled hospital admissions in Poland were suspended throughout most of 2020 [27]. Moreover, numerous outpatient clinics suspended in-person visits and medical services were provided using teleconsultations [28]. The COVID-19 pandemic limited access to regular check-ups by diabetic patients [29,30], increased the number of diagnosed diabetic ketoacidosis cases, and led to a reduction in hospitalization time [26]. However, nationwide data on the access to healthcare among diabetic patients are very limited. Detailed analysis of hospitalization related to diabetes mellitus in Poland before and during COVID-19 is crucial to assess the potential impact of the pandemic on the health status of diabetic patients as well as to prove evidence-based public health interventions targeted to patients with diabetes.

Therefore, this study aimed to present an epidemiological analysis of hospitalization related to diabetes mellitus in Poland between 2014 and 2020, with particular emphasis on (1) trends in diabetes-related hospital admissions; (2) duration of hospitalization; and (3) in-hospital mortality.

## 2. Materials and Methods

### 2.1. Data Source

This epidemiological analysis is based on data from the Polish National Hospital Register carried out by the National Institute of Public Health–National Institute of Hygiene (NIPH—NIH) within the population-based hospital morbidity study [31]. Discharge reports are submitted to NIPH—NIH by the public and private hospitals (except the psychiatric units) from all administrative regions in Poland. Data are reported using the template specified by Polish law. All discharge reports are anonymous and contain information on age, gender, date of admission, date of discharge, primary diagnosis (cause of hospitalization), secondary diagnosis (co-existing diseases), the outcome of hospitalization, and medical procedures provided during the hospitalization [32]. Data on the patients’ medical conditions were filled by the physicians and based on ICD-10 codes (10th revision of the International Statistical Classification of Diseases and Related Health Problems) [33].

In this study, data on all patients for whom diabetes was the cause of hospitalization were included in the analysis. Diabetes-related hospital admissions were identified using the following ICD-10 codes: E10 (insulin-dependent diabetes mellitus); E11 (non-insulin-dependent diabetes mellitus); E13 (other specified diabetes mellitus); E14 (unspecified diabetes mellitus) [33].

This study was carried out following the principles expressed in the Declaration of Helsinki and approved by the Ethical Review Board at the Centre of Postgraduate Medical Education, Warsaw, Poland (no. 70/2022).

### 2.2. Statistical Analysis

All analyses were performed with IBM SPSS ver. 28. Descriptive statistics were used.

Diabetes-related hospital admissions were presented by crude numbers as well as hospitalization rate per 100,000 inhabitants (number of diabetes-related hospitalizations per year by age groups). Data on the Polish population was based on the Demographic Database provided by the Central Statistical Office, Warsaw, Poland [34].

Duration of hospitalization was presented by means along with the standard deviation (SD). The in-hospital mortality rate was calculated as the percentage of fatal cases out of diabetes-related hospital admissions in particular groups. Cross-tabulations and chi-squared tests were used to compare categorical variables.

Data were analyzed separately for type 1 (ICD-10 code: E10) and type 2 diabetes (ICD-10 code: E11). Percentage differences in hospitalization trends before (2019) and during the COVID-19 pandemic (2020) were calculated.

## 3. Results

### 3.1. Trends in Diabetes-Related Hospitalizations in Poland between 2014 and 2020

The number of diabetes-related hospitalizations varied from 76,220 in 2016 to 45,159 in 2020 (Table 1 and Appendix A). Regardless of the analyzed year, more than half of the patients hospitalized with diabetes (ICD-10 codes: E10, E11, E13, E14) had type 2 diabetes (ICD-10 code: E11). Since 2017, the number of people hospitalized for diabetes has decreased every year. However, in the period 2019–2020 the overall number of diabetes-related hospitalizations decreased by over a third (Appendix A). Among patients with type 1 diabetes, hospitalization rate per 100,000 has decreased from 74.6 in 2019 to 53.0 in 2020 (percentage change: −28.9%). An even greater drop was observed among patients with type 2 diabetes: from 99.4 in 2019 to 61.6 in 2020 (percentage change: −38%).

Among patients with both type 1 and type 2 diabetes (Table 1), a decrease in hospitalization rate was higher among females than males (−31.6% vs. −26.7% and −40.9% vs. −35.2% respectively; *p* < 0.001).

### 3.2. Type 1 Diabetes-Related Hospitalizations by Gender and Age

Throughout the analyzed period (2014–2020), the overall hospitalization rate due to type 1 diabetes was higher among males than females (Table 1). Most of the patients hospitalized with type 1 diabetes were aged 10–19 years (Figure 1 and Appendix A). Two peaks in hospitalization rate due to type 1 diabetes were observed: the first among individuals aged 10–19, and the second among individuals aged 60 and over (Figure 1 and Appendix A). Since 2014, the type 1 diabetes hospitalization rate in the oldest age groups (70–79 years and 80 years and over) had constantly decreased. Between 2015 and 2016, a markable increase in the hospitalization rate was observed among individuals aged 10–19 years (from 121.6 to 142.0 per 100,000 inhabitants). During the COVID-19 pandemic (2020), the highest decrease in type 1 diabetes hospitalization rate was observed among individuals aged 30–39 years (−38.1%) or 20–29 years (−37.3%). Among patients aged 60 and over, the hospitalization rate dropped by a quarter. The lowest decrease was observed in the youngest age group (0–9 years; −22.9%). Except for the group aged 0–9 years, a higher decrease in hospitalization rate was observed among females than males (Figure 1 and Appendix A).

### 3.3. Type 2 Diabetes-Related Hospitalizations by Gender and Age

Since 2015, the overall hospitalization rate due to type 2 diabetes was higher among males than females (Table 1). The type 2 diabetes hospitalization rate increased with the age, with a markable peak after 40 years of age (Figure 2 and Appendix A). In the age groups 40–49 and 50–59, the hospitalization rate was more than two times higher among males than females. During the COVID-19 pandemic (2020), the highest decrease in type 2 diabetes-hospitalization rate was observed among individuals aged 40–49 years (−41.2%) and those aged 80 years and over (−40.3%). Among patients aged 50–79, the hospitalization rate has dropped by over a third. The lowest decrease was observed in the youngest age group (0–9 years; −7.8%). In all age groups, a higher decrease in hospitalization rate was observed among females than males (Figure 2 and Appendix A).

### 3.4. The Average Duration of Hospitalization

The average duration of diabetes-related hospitalization exceeded 7 days (Table 2 and Table 3). In both groups, the average duration of hospitalization differed by age group. Among patients hospitalized with type 1 diabetes, the average duration of hospitalization markedly increased after 60 years of age (Table 2). In general, the average duration of hospitalization due to type 1 diabetes was higher among males than females. Between 2019 and 2020, the average duration of hospitalization increased from 7.2 ± 8.7 days to 7.7 ± 11.3 (Table 4). Among patients hospitalized with type 2 diabetes, the average duration of hospitalization steadily increased with the age (Table 3). In most age groups, the average duration of hospitalization due to type 2 diabetes was higher among males than females. Between 2019 and 2020, the average duration of hospitalization increased from 7.7 ± 7.5 days to 8.1 ± 9.9 (Table 3).

### 3.5. In-Hospital Mortality Rate

There were significant differences in in-hospital mortality rate both among patients with type 1 and type 2 diabetes, depending on the analyzed period (Table 4 and Table 5). In general, the in-hospital mortality rate increased with the age and was higher among females (regardless of the type of diabetes). Among the patients hospitalized with type 1 diabetes, the in-hospital mortality rate markedly increased after 70 years of age (Table 4). Among the patients hospitalized with type 2 diabetes, the in-hospital mortality rate noticeably increased after 80 years of age (Table 5). When compared to 2019, in 2020 in-hospital mortality rate has increased by 66.7% (60.0% among males and 65.2% among females) among patients hospitalized with type 1 diabetes and by 48.5% (55.2% among females and 42.1% among males) among patients hospitalized with type 2 diabetes. The percentage change in in-hospital mortality rate differed by age group both in a group of patients with type 1 diabetes (Table 4) and type 2 diabetes (Table 5).

## 4. Discussion

This study provided a comprehensive epidemiological analysis of hospitalization related to diabetes mellitus in Poland over the last 7 years (2014–2020). Findings from this study allowed us to assess the impact of the COVID-19 pandemic on diabetes-related hospitalizations in different age groups. Between 2019 and 2020, the hospitalization rate per 100,000 inhabitants decreased by 28.9% among patients with type 1 diabetes and by 38% among patients with type 2 diabetes. During the COVID-19 pandemic, an increase in diabetes-related in-hospital mortality rate was observed. Moreover, in this study markable differences in hospitalization rate, duration of hospitalization, as well as in-hospital mortality rate by gender, were observed, which reveal health inequalities.

Most diabetic patients may be treated in ambulatory care [19,20,21]. However, effective ambulatory care requires high therapeutic compliance among diabetic patients [35]. Regular blood sugar monitoring, healthy lifestyle, diet control as well as taking medications regularly requires high levels of acceptance among the patients and depends on the individual’s health literacy level [36,37]. Most diabetes-related hospitalizations are caused by severe dysglycemia [35]. Among patients with type 2 diabetes, most of the hospitalizations result from low compliance or the existence of diseases that may have an impact on blood sugar levels or general health status. In Poland, hospitalizations due to type 1 diabetes among children are the result of national guidelines, i.e., that all children newly diagnosed with type 1 diabetes should be hospitalized in a specialist diabetes unit [25]. Due to the organization of the healthcare systems in Poland, general practitioner who diagnosed children with diabetes refers their patients to such diabetes units [25]. During these hospitalizations, minors with type 1 have a health screening to assess their current health condition and adjust their treatment. A basic education to patients and their parents in the field of diabetic care is also provided.

Out of all diabetes-related hospitalizations analyzed in this study, approximately 52–55% were among patients with type 2 diabetes, and 41–45% were among patients with type 1 diabetes. The hospitalization rate due to type 1 diabetes in Poland is higher than in other EU countries due to the organization of the healthcare in Poland, which obligates physicians to hospitalize new type 1 diabetes patients [25]. This study confirmed that the COVID-19 pandemic significantly limited access to healthcare among patients with diabetes. In a group of patients with type 1 diabetes, the hospitalization rate per 100,000 decreased by 28.9%. Even a greater decrease was observed among patients with type 2 diabetes (by 38%). In this study, females were more likely to expect limited access to diabetic care during the COVID-19 pandemic, as a higher drop in the hospitalization rate was observed among males than females, regardless of the type of diabetes. Moreover, the percentage change in hospitalization rate per 100,000 also varied by age. These findings point to markable health inequality in access to diabetic care during the COVID-19 pandemic by gender and age. Such differences in access to medical care, combined with proven socioeconomic inequalities in the incidence and prevalence of type 2 diabetes [38], may lead to a growing number of diabetes-related complications among females and those age groups, where access to healthcare was significantly limited, thus amplifying existing health inequalities.

The hospitalization rate due to type 1 diabetes was higher among males, which may result from the fact that type 1 diabetes is more prevalent among males. Among the patients hospitalized with type 1 diabetes, two peaks in hospitalization rate were observed—among those aged 10–19 years and patients over 60 years of age. Although type 1 diabetes can be diagnosed at any age, most cases are commonly diagnosed between the ages of 10–15 years [39,40]. We can hypothesize that a higher hospitalization rate among patients aged 10–19 years results from hospitalizations due to newly diagnosed diabetes. The peak in hospitalization rate among individuals over 60 years of age may result from the health status of this age group, in particular comorbidities as well as diabetes-related complications. Age of 60 years and over was identified as an independent risk factor for diabetes-related complications [41]. In this study, the highest drop in type 1 diabetes-related hospitalization rate was observed among young adults aged 20–39 years and exceeded 37 percentage points. The lowest change in the hospitalization rate was observed among the youngest group (0–9 years) which may result from the fact that children are at higher risk of sudden deterioration of their health status, so physicians may tend to maintain hospital admissions in this group.

Obesity is the most significant risk factor for the development of type 2 diabetes [42]. In most populations, the prevalence of obesity in adults is greater for women than for men [43]. However, national data showed that in Poland, the prevalence of overweight or obesity is higher among men (68.9% vs. 48.2%) [44]. In this study, the hospitalization rate due to type 2 diabetes was higher among men, which may result from the sex differences in the prevalence of overweight/obesity in Poland [44]. The hospitalization rate due to type 2 diabetes increases with the age. The hospitalization rate among patients aged 40–59 years was almost two times higher among men than women which may reflect the health status of the polish population [45]. Moreover, during the COVID-19 pandemic in almost all age groups, the highest drop in the hospitalization rate was observed among females. This finding points to health inequalities also in the case of type 2 diabetes. Each year, approximately 150–180 minors were hospitalized due to type 2 diabetes. We can hypothesize that type 2 diabetes among minors may result from the social changes and unhealthy lifestyles that led to the obesity epidemic among children [46].

In this study, the duration of hospitalization was also analyzed. The average duration of hospitalization was approximately 7 days, wherein the duration of hospitalization was higher among males. Moreover, the duration of hospitalization differed between the age groups. Among patients with type 1 diabetes, a relatively long hospitalization was observed among patients aged 0–9 years. We can hypothesize that this age group is of particular care due to the early onset of the disease, which has a markable impact on the prognosis and course of the disease [47]. In the Polish health system, the time spent in the hospital is also used to educate the patient’s careers on diabetes management, as access to such education in outpatient settings is limited [48]. Among patients with type 2 diabetes, the duration of hospitalization increased with the age. Age is a risk factor for numerous diseases, so we can hypothesize that older patients are at higher risk of diabetes-related complications and comorbidities that may extend the length of hospitalization [49]. During the COVID-19 pandemic, the mean duration of hospitalization increased both among patients with type 1 and type 2 diabetes. Diabetes is a risk factor for severe COVID-19, so we can hypothesize that stabilization of the clinical condition of a patient hospitalized due to diabetes was an important priority for doctors, which influenced the duration of hospitalization [32]. In this study, a wide range of lengths of hospitalization was observed (as expressed in the values of standard deviations) that may indicate markable differences in diabetic care by hospitals and geographic regions.

In this study, the in-hospital mortality rate increased with the age and was higher among females (both for type 1 and type 2 diabetes). Moreover, there were differences in in-hospital mortality rate by study year. However, during the COVID-19 pandemic (2019–2020) in-hospital mortality rate among patients with type 1 diabetes increased by 66.7% and by 48.5% among patients with type 2 diabetes. This finding revealed that the COVID-19 pandemic had a negative impact both on access to healthcare and the quality of care. A markable increase in the in-hospital mortality rate during the COVID-19 pandemic may result from the fact that the number of healthcare workers was limited, as many of them worked in COVID-19 dedicated centers [50]. Further research is needed to identify the reasons for such a high increase in the in-hospital mortality rate among patients hospitalized with diabetes in Poland.

This study has several practical implications for healthcare workers and policymakers. This study provided scientific evidence on the impact of the COVID-19 pandemic on diabetic care in Poland. A markable decrease in diabetes-related hospitalizations and an increase in in-hospital mortality rate may lead to the deterioration of the health status of the population. Moreover, a drop in the hospitalization rate may lead to a rise in the number of diabetes-related hospitalization in the future, further boosting the diabetes burden. In this study, health inequalities in diabetic care were revealed. The hospitalization rate (both due to type 1 and type 2 diabetes) was lower among females, wherein the in-hospital mortality rate was higher. Policymakers should identify and remove barriers to access to healthcare based on gender or age.

This study has some limitations. First, this is a retrospective analysis of the national registry dataset, so data are limited to the one specific template used in discharge reports collected by the NIPH—NIH. Second, only data on hospital admissions were included in the analysis. Data on outpatients’ visits were not included in the registry, so the full scope of diabetes care in Poland was not analyzed. Nevertheless, this is the most comprehensive analysis of hospitalizations due to diabetes during the past 7 years. Third, the geographical distribution of diabetic care centers was not analyzed in this study. A relatively high number of older patients (80 years and over) hospitalized with type 1 diabetes may suggest some coding/diagnosis bias. Nevertheless, we used a nationwide representative registry led by the designated public governmental institution. Further actions are needed to regularly monitor and improve the data quality available in the public medical registries.

## 5. Conclusions

During the first year of the COVID-19 pandemic in Poland, a markable decrease in diabetes-related hospitalization rates and an increase in in-hospital mortality rate have been observed. Markable differences in hospital admissions, duration of hospitalization, and in-hospital mortality rate by gender and age observed in this study point to health inequalities. Epidemiological analysis of nationwide data is crucial to assess the quality of diabetic care as well as to identify potential differences in diabetic care by gender, age, or place of residence.

## Figures and Tables

**Figure 1 ijerph-19-10030-f001:**
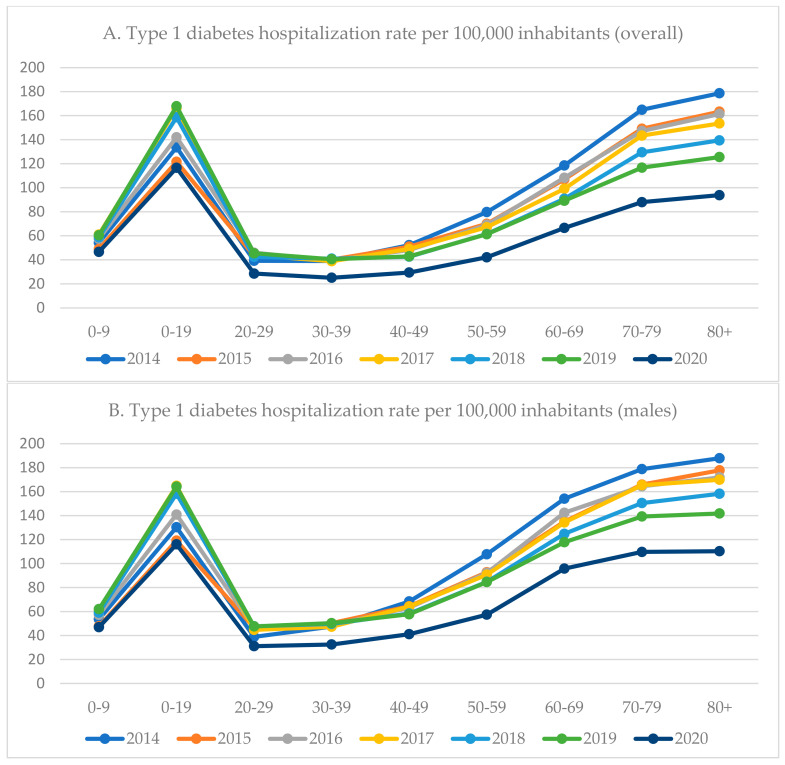
Hospitalization trends among patients with type 1 diabetes in Poland between 2014–2019 (hospitalization rate per 100,000 inhabitants) by age groups.

**Figure 2 ijerph-19-10030-f002:**
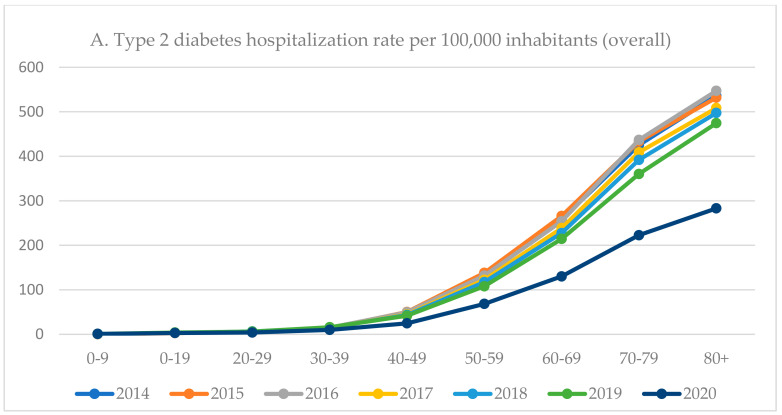
Hospitalization trends among patients with type 2 diabetes in Poland, 2014–2019 (hospitalization rate per 100,000 inhabitants) by age groups.

**Table 1 ijerph-19-10030-t001:** Diabetes-related hospitalizations in Poland between 2014–2019.

ICD-10	2014 *n* = 75,476			2015 *n* = 74,826			2016 *n* = 76,220			2017 *n* = 74,670			2018 *n* = 72,188		
	T	M	F	T	M	F	T	M	F	T	M	F	T	M	F
	Hospitalization Rate (Number of Diabetes-Related Hospitalizations) per 100,000 Inhabitants														
Overall	196.2	205.5	187.3	194.7	206.4	183.7	198.3	211.4	186.0	194.3	209.9	179.7	187.9	205.2	171.7
E10	80.4	89.2	72.1	75.2	83.7	67.2	78.1	87.5	69.3	79.0	89.4	69.2	74.4	85.9	63.6
E11	107.3	106.0	108.4	110.8	112.1	109.6	111.4	113.0	110.0	106.0	108.7	103.3	103.9	107.4	100.6
E13	4.5	5.5	3.5	4.7	5.9	3.6	4.6	5.8	3.4	4.5	5.8	3.3	4.3	5.6	3.1
E14	4.0	4.8	3.3	4.0	4.8	3.2	4.2	5.1	3.4	4.9	5.9	3.8	5.3	6.2	4.4
**ICD-10**	**2019 *n* = 68,906**				**2020 *n* = 45,159**					**Percentage Difference 2019–2020**					
	**T**		**M**	**F**	**T**		**M**	**F**		**Overall**		**Males**		**Females**	** *p* **
Overall	179.5		196.4	163.7	118.0		134.3	102.7		−34.2		−31.6		−37.2	<0.001
E10	74.6		85.4	64.4	53.0		62.5	44.1		−28.9		−26.7		−31.6	<0.001
E11	99.4		104.1	95.0	61.6		67.5	56.1		−38.0		−35.2		−40.9	<0.001
E13	0.5		0.6	0.3	0.3		0.4	0.2		−38.0		−38.4		−37.1	0.9
E14	5.2		6.3	4.0	3.1		3.9	2.4		−39.0		−37.7		−41.0	0.5

Abbreviations: T—total; M—males; F—females; *p*—the result of the chi-squared tests (gender differences in the number of patients hospitalized with diabetes 2019 vs. 2020).

**Table 2 ijerph-19-10030-t002:** The average duration of hospitalization (in days) among patients hospitalized with type 1 diabetes, Poland, 2014–2020.

Age Group	2014 *n* = 30,931						2015 *n* = 28,890						2016 *n* = 30,027						2017 *n* = 30,351					
	Overall		Males		Females		Overall		Males		Females		Overall		Males		Females		Overall		Males		Females	
	X	SD	X	SD	X	SD	X	SD	X	SD	X	SD	X	SD	X	SD	X	SD	X	SD	X	SD	X	SD
Total	7.6	9.7	7.7	8.5	7.4	10.9	7.6	8.1	7.8	8.8	7.4	7.3	7.4	8.5	7.7	9.2	7.0	7.4	7.4	9.5	7.7	9.2	7.0	9.9
0–9	5.2	5.0	5.1	4.8	5.2	5.2	5.3	5.2	5.2	5.0	5.5	5.3	5.2	6.3	5.3	7.3	5.1	5.0	4.9	4.8	5.0	4.8	4.8	4.7
10–19	4.4	3.9	4.5	4.1	4.4	3.8	4.5	3.9	4.5	3.9	4.5	3.7	4.4	3.8	4.3	3.8	4.4	3.8	4.0	3.7	3.9	3.7	4.0	3.7
20–29	5.6	3.7	5.6	3.5	5.6	4.0	5.6	6.8	5.8	4.3	5.4	3.4	5.2	3.3	5.3	3.5	5.1	3.1	5.5	4.5	5.5	4.1	5.4	4.9
30–39	6.3	5.9	6.6	6.3	6.0	5.4	6.4	8.1	6.5	5.9	6.3	8.2	6.2	5.4	6.5	5.8	5.7	4.5	6.4	6.7	6.5	6.3	6.1	7.3
40–49	7.3	6.6	7.5	7.2	7.0	5.1	7.8	9.2	8.0	8.4	7.4	7.3	7.4	8.0	7.6	7.1	7.1	9.6	7.7	7.7	7.8	7.4	7.6	8.3
50–59	8.5	8.9	8.8	9.6	8.0	7.5	8.9	9.8	9.3	9.9	8.3	7.7	8.8	10.5	9.5	11.3	7.7	8.6	9.3	10.5	9.8	11.4	8.2	8.2
60–69	9.6	11.1	9.8	11.0	9.4	11.3	9.7	9.2	9.9	10.5	9.3	8.7	9.6	10.2	10.2	11.4	8.7	8.2	9.8	10.7	10.5	11.7	8.8	8.8
70–79	9.9	15.5	10.3	10.6	9.7	18.3	9.4	9.8	10.0	10.5	8.9	8.1	9.4	9.8	9.8	10.1	9.0	9.4	9.6	10.3	10.3	11.4	9.1	9.3
80+	9.2	12.7	9.0	8.0	9.3	14.4	9.3	8.1	9.8	12.8	9.1	8.0	9.2	10.7	9.6	14.5	9.0	8.2	10.0	17.8	10.0	11.5	10.0	20.3
**Age Group**	**2018 *n* = 28,584**									**2019 *n* = 28,617**								**2020 *n* = 20,282**						
	**Overall**			**Males**			**Females**			**Overall**			**Males**		**Females**			**Overall**		**Males**		**Females**		
	**X**	**SD**		**X**	**SD**		**X**	**SD**		**X**	**SD**		**X**	**SD**	**X**	**SD**		**X**	**SD**	**X**	**SD**	**X**	**SD**	
Total	7.6	8.8		7.8	9.6		6.9	7.6		7.2	8.7		7.5	9.4	6.8	7.8		7.7	11.3	8.0	11.3	7.2	11.2	
0–9	5.1	5.1		5.3	5.2		4.9	5.0		5.0	5.0		4.9	5.1	5.0	4.9		6.0	7.1	6.1	8.5	5.9	5.3	
10–19	4.1	3.7		4.1	3.7		4.1	3.7		3.9	3.8		3.8	3.6	4.0	4.0		4.3	4.5	4.2	4.1	4.3	4.9	
20–29	5.3	4.0		5.3	3.6		5.3	4.5		5.3	3.8		5.4	3.5	5.2	4.1		5.4	3.9	5.5	4.2	5.4	3.4	
30–39	6.3	6.4		6.7	7.3		5.8	4.4		6.2	5.5		6.4	5.7	5.8	5.0		6.6	7.3	6.7	7.2	6.4	7.5	
40–49	7.7	7.8		8.0	8.3		6.8	6.5		7.4	7.3		7.6	7.6	7.0	6.6		7.8	8.0	7.8	7.7	7.6	8.6	
50–59	8.9	9.2		9.4	9.8		8.0	7.6		9.2	12.1		9.6	13.4	8.3	8.7		9.6	11.7	10.1	12.4	8.4	10.0	
60–69	10.0	11.8		10.8	12.9		8.7	9.4		9.9	11.1		10.4	12.0	9.0	9.7		10.4	15.5	11.0	17.4	9.1	10.9	
70–79	10.3	11.5		11.2	12.3		9.4	10.7		9.7	10.7		10.2	11.1	9.3	10.3		10.4	17.6	10.3	12.3	10.4	21.8	
80+	9.6	10.6		9.8	13.1		9.4	9.0		9.5	10.6		10.1	11.6	9.2	9.9		9.5	10.3	9.9	11.9	9.2	9.2	

**Table 3 ijerph-19-10030-t003:** The average duration of hospitalization (in days) among patients hospitalized with type 2 diabetes, Poland, 2014–2020.

Age Group	2014 *n* = 41,275						2015 *n* = 42,599						2016 *n* = 42,821						2017 *n* = 40,722					
	Overall		Males		Females		Overall		Males		Females		Overall		Males		Females		Overall		Males		Females	
	X	SD	X	SD	X	SD	X	SD	X	SD	X	SD	X	SD	X	SD	X	SD	X	SD	X	SD	X	SD
Total	7.6	7.1	7.6	6.9	7.7	7.3	7.7	7.7	7.7	7.5	7.7	7.9	7.7	7.7	7.7	8.6	7.6	6.7	7.8	7.9	7.7	7.7	7.8	8.1
0–9	3.4	6.1	5.3	8.2	1.6	2.3	4.1	5.2	3.4	5.3	4.6	5.5	7.4	11.8	9.0	14.3	5.0	5.8	5.1	7.7	6.4	10.1	3.6	3.9
10–19	4.4	3.5	4.4	3.9	4.5	3.2	4.2	4.0	4.4	3.4	4.0	3.7	4.3	4.1	4.1	4.5	4.5	3.6	3.6	3.5	3.0	2.9	4.0	3.8
20–29	5.1	3.5	5.2	3.5	5.1	3.5	5.3	3.4	5.6	5.5	4.9	3.3	5.3	3.3	5.3	3.4	5.3	3.2	5.3	3.3	5.3	3.5	5.2	3.0
30–39	5.6	3.7	5.7	3.8	5.5	3.5	6.1	5.8	6.2	6.2	5.7	4.8	6.1	5.4	6.2	5.4	5.8	5.5	6.0	5.0	6.0	5.2	5.9	4.3
40–49	6.5	5.1	6.5	5.1	6.5	4.9	6.6	5.8	6.6	6.9	6.4	4.6	6.6	6.3	6.8	6.9	6.0	4.0	6.7	7.0	6.8	7.2	6.3	6.6
50–59	7.2	6.3	7.3	6.6	7.1	6.0	7.2	6.8	7.3	8.6	7.1	6.7	7.1	7.0	7.4	7.5	6.7	6.0	7.3	6.9	7.4	6.8	7.0	7.2
60–69	7.5	7.0	7.7	7.8	7.2	6.1	7.8	7.9	7.9	7.2	7.6	6.8	7.6	9.1	7.8	10.3	7.4	7.1	7.7	7.8	7.8	8.4	7.5	6.8
70–79	8.1	7.3	8.2	7.3	8.0	7.4	8.0	6.6	8.2	7.3	7.9	6.3	8.0	7.5	8.4	8.6	7.8	6.6	8.0	8.9	8.3	8.3	7.9	9.3
80+	8.3	8.3	8.3	6.5	8.4	8.9	8.3	9.9	8.4	7.5	8.3	10.8	8.4	7.0	8.4	7.4	8.4	6.8	8.5	8.0	8.4	7.0	8.6	8.4
**Age Group**	**2018 *n* = 39,904**									**2019 *n* = 38,138**								**2020 *n* = 23,568**						
	**Overall**			**Males**			**Females**			**Overall**			**Males**		**Females**			**Overall**		**Males**		**Females**		
	**X**	**SD**		**X**	**SD**		**X**	**SD**		**X**	**SD**		**X**	**SD**	**X**	**SD**		**X**	**SD**	**X**	**SD**	**X**	**SD**	
Total	7.9	8.8		8.0	8.5		7.8	9.0		7.7	7.5		7.8	7.8	7.6	7.2		8.1	9.9	8.2	9.2	8.0	10.6	
0–9	5.4	5.3		5.6	5.7		5.2	5.2		5.0	5.4		4.5	6.1	5.3	4.9		3.7	4.8	4.1	4.8	3.2	4.9	
10–19	3.9	3.3		4.2	3.6		3.4	2.9		4.2	3.7		4.8	4.2	3.8	3.2		4.4	4.1	4.3	4.5	4.5	3.8	
20–29	5.2	4.5		5.8	5.3		4.5	3.1		5.1	3.4		5.2	3.8	4.8	2.8		5.4	5.1	5.8	5.8	4.9	3.8	
30–39	6.4	6.6		6.3	6.2		6.5	7.6		5.8	5.8		5.7	4.2	6.2	8.3		6.0	5.6	6.1	5.0	5.9	7.2	
40–49	6.5	5.8		6.6	5.9		6.3	5.6		6.5	6.2		6.6	6.6	6.1	5.0		7.1	7.7	7.3	7.6	6.7	8.1	
50–59	7.5	7.5		7.8	8.3		6.9	5.6		7.5	8.4		7.9	9.2	6.9	6.5		7.9	8.9	8.2	9.7	7.0	6.8	
60–69	7.9	8.3		8.1	8.8		7.5	7.4		7.7	7.6		7.9	8.0	7.4	7.0		8.1	8.6	8.3	9.1	7.7	7.6	
70–79	8.1	8.6		8.4	8.6		7.9	8.6		8.0	8.1		8.3	8.0	7.8	8.1		8.4	9.2	8.7	9.9	8.2	8.4	
80+	8.7	11.1		8.8	9.7		8.6	11.7		8.2	6.8		8.2	6.7	8.3	6.9		8.8	13.6	8.7	9.0	8.8	15.1	

**Table 4 ijerph-19-10030-t004:** In-hospital mortality rate (%) among patients hospitalized with type 1 diabetes, Poland, 2014–2020.

Age Group	2014 *n* = 30,931			2015 *n* = 28,890			2016 *n* = 30,027			2017 *n* = 30,351			2018 *n* = 28,584			2019 *n* = 28,617			2020 *n* = 20,282			Percentage Difference 2019–2020			
	T	M	F	T	M	F	T	M	F	T	M	F	T	M	F	T	M	F	T	M	F	T	M	F	*p*
Overall	1.9	1.7	2.1	2.1	1.9	2.4	2.2	2.0	2.5	2.2	2.0	2.4	2.4	2.2	2.6	2.1	2.0	2.3	3.5	3.2	3.8	66.7	60.0	65.2	<0.001
0–9	0.0	0.0	0.0	0.0	0.0	0.0	0.0	0.0	0.0	0.0	0.0	0.0	0.0	0.0	0.0	0.0	0.0	0.0	0.0	0.0	0.0	N/A	N/A	N/A	N/A
10–19	0.0	0.0	0.0	0.0	0.0	0.0	0.0	0.0	0.0	0.0	0.0	0.0	0.0	0.0	0.0	0.0	0.0	0.0	0.0	0.0	0.0	N/A	N/A	N/A	N/A
20–29	0.1	0.1	0.1	0.1	0.1	0.1	0.1	0.0	0.2	0.2	0.3	0.2	0.2	0.1	0.2	0.1	0.2	0.1	0.2	0.3	0.0	N/A	N/A	N/A	N/A
30–39	0.6	0.7	0.3	0.4	0.5	0.1	0.5	0.6	0.2	0.4	0.6	0.1	0.8	1.0	0.4	0.6	0.8	0.3	1.1	1.6	0.2	N/A	N/A	N/A	N/A
40–49	0.6	0.5	0.8	1.3	1.6	0.9	0.7	0.8	0.5	1.0	1.5	0.1	0.9	1.1	0.6	1.0	1.3	0.4	1.8	2.1	1.0	80.0	61.5	150.0	0.03
50–59	1.2	1.4	0.7	1.6	1.7	1.5	1.9	1.8	2.0	1.5	1.2	2.0	1.7	1.7	1.7	1.9	2.2	1.3	2.7	3.1	2.0	42.1	40.9	53.8	0.052
60–69	2.2	2.6	1.6	2.6	2.3	2.9	2.8	2.9	2.6	3.0	3.2	2.6	3.2	3.0	3.6	3.0	2.7	3.5	4.5	4.7	4.1	50.0	74.1	17.1	<0.001
70–79	4.4	4.6	4.2	4.7	4.7	4.8	4.9	4.7	5.2	4.5	5.2	3.9	5.0	5.8	4.3	5.1	5.2	4.9	8.8	8.0	9.6	72.5	53.8	95.9	<0.001
80+	7.7	6.9	8.1	8.3	8.8	8.0	9.3	10.1	8.9	10.0	9.5	10.3	11.3	12.5	10.6	9.7	9.8	9.7	14.1	13.2	14.7	45.4	34.7	51.5	<0.001

Abbreviations: T—total; M—males; F—females; N/A—percentage difference was not calculated due to limited sample size; *p*—the result of the chi-squared tests (gender differences in the in-hospital mortality rate 2019 vs. 2020).

**Table 5 ijerph-19-10030-t005:** In-hospital mortality rate among patients hospitalized with type 2 diabetes, Poland, 2014–2020.

Age Group	2014 *n* = 41,275			2015 *n* = 42,599			2016 *n* = 42,821			2017 *n* = 40,722			2018 *n* = 39,904			2019 *n* = 38,138			2020 *n* = 23,568			Percentage Difference 2019–2020			
	T	M	F	T	M	F	T	M	F	T	M	F	T	M	F	T	M	F	T	M	F	T	M	F	*p*
Overall	2.9	2.4	3.3	2.9	2.4	3.3	3.2	2.8	3.5	2.9	2.4	3.4	3.3	2.8	3.8	3.3	2.9	3.8	4.9	4.5	5.4	48.5	55.2	42.1	<0.001
0–9	0.0	0.0	0.0	0.0	0.0	0.0	0.0	0.0	0.0	0.0	0.0	0.0	0.0	0.0	0.0	0.0	0.0	0.0	0.0	0.0	0.0	N/A	N/A	N/A	N/A
10–19	0.0	0.0	0.0	0.0	0.0	0.0	0.0	0.0	0.0	0.0	0.0	0.0	0.0	0.0	0.0	0.0	0.0	0.0	0.0	0.0	0.0	N/A	N/A	N/A	N/A
20–29	0.7	0.6	1.0	0.0	0.0	0.0	0.4	0.6	0.0	0.3	0.0	0.8	0.3	0.6	0.0	0.0	0.0	0.0	0.5	0.0	1.4	N/A	N/A	N/A	N/A
30–39	0.4	0.5	0.0	0.4	0.3	0.7	0.3	0.4	0.0	0.1	0.2	0.0	0.6	0.6	0.8	0.7	0.6	1.0	1.8	1.8	1.9	N/A	N/A	N/A	N/A
40–49	0.2	0.3	0.1	0.8	0.8	0.8	0.9	1.1	0.4	0.8	0.8	0.8	1.0	1.0	1.2	1.0	1.0	0.9	1.1	1.2	0.5	10.0	20.0	−44.4	0.9
50–59	1.1	1.2	1.1	1.1	1.1	1.2	1.1	1.1	1.2	0.9	0.8	0.9	1.3	1.3	1.2	1.1	1.2	0.9	2.0	1.8	2.3	81.8	50.0	155.6	0.001
60–69	1.6	1.7	1.6	1.5	1.4	1.6	2.0	2.1	1.8	1.7	1.6	1.8	2.1	2.1	2.0	2.0	2.2	1.9	3.5	3.5	3.3	75.0	59.1	73.7	<0.001
70–79	3.3	3.6	3.1	3.2	3.6	3.0	3.4	3.9	3.1	2.9	3.2	2.6	3.2	3.7	2.8	3.5	3.7	3.4	5.3	5.9	4.7	51.4	59.5	38.2	<0.001
80+	6.8	6.9	6.8	7.1	7.7	6.8	7.2	7.8	6.9	7.1	7.3	7.0	7.5	6.9	7.8	7.6	7.8	7.5	10.2	11.8	9.4	34.2	51.3	25.3	<0.001

Abbreviations: T—total; M—males; F—females; N/A—percentage difference was not calculated due to limited sample size; *p*—the result of the chi-squared tests (gender differences in the in-hospital mortality rate 2019 vs. 2020).

## Data Availability

Data are available on reasonable request. The dataset used to conduct the analyses is available from corresponding author on reasonable request.

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
