# Peer review of "Epidemiological Analysis of Diabetes-Related Hospitalization in Poland before and during the COVID-19 Pandemic, 2014–2020"

_ijerph, 2022, doi:10.3390/ijerph191610030_

Round 1
Reviewer 1 Report
In this manuscript authors analyzed the changes in diabetes-related hospital admissions in the period between 2014 and 2020, therefore before and during the COVID-19 pandemic (before the beginning of Covid-19 vaccination). Thestudy was a retrospective analysis based on national registry dataset of hospital discharge reports on diabetes-related hospitalizations. They found that the hospitalization rate per 100,000 has decreased both for type 1 diabetes (percentage change: -28.9%) and type 2 diabetes (percentage change: -38%).
Although it is likely that during Covid-19 pandemic the hospitalization for diabetes related complications (both acute and chronic) reduced, the data presented in this manuscript extremely implausible. In fact, since the poland population consists of 38 milions of individuals, and the polish prevalence of diabetes is about 7%, this results in a approximate number of 2.700.000 diabetes individuals. 95% of these are Type 2 diabetic patients (2.565.000), the rest with T1D, autoimmune disease, (about 135.000 individuals). Even considering the rate of unknown diabetes, that is clearly higher for T2D while highly unlikely for T1D, the data reported by authors are not credible. In fact, how is possible that a disease almost 100 times more common than another has more or less the same hospitalization rate? Moreover, acute and chronic Type 2 diabetes complications such as glycemic unbalanced, hyperosmolar coma as well as infections, and all the other diabetes related vascular complications account for the most common causes of hospitalizations. Therefore, I think that the data acquired from the authors have a big coding problem. By the way, this issue is confirmed looking at Table 2 where is stated that between 2014 and 2019 more than 2000 Type 1 diabetes patients aged 80 and older were hospitalyzed every year. I would remind that life expectancy of individuals with type 1 diabetes is about 8-10 years less than general population (Cardiovasc Endoc Metab 2020. Doi: 10.1097/XCE.0000000000000210), therefore to came across in over 80 individuals with T1D is really uncommon, to record thousands of very old people with T1D is very unlikely.
Finally, E12 stands for malnutrition-related diabetes mellitus that is not a type of diabetes, therefore it should be removed forom Table 1
Author Response
Thank you very much for all the valuable comments that allow the Authors to increase the scientific soundness of our paper.
COMMENT #1: In this manuscript authors analyzed the changes in diabetes-related hospital admissions in the period between 2014 and 2020, therefore before and during the COVID-19 pandemic (before the beginning of Covid-19 vaccination). The study was a retrospective analysis based on national registry dataset of hospital discharge reports on diabetes-related hospitalizations. They found that the hospitalization rate per 100,000 has decreased both for type 1 diabetes (percentage change: -28.9%) and type 2 diabetes (percentage change: -38%).
RESPONSE #1: Thank you for this comment. Our major goal was to present a comprehensive characteristic of patients hospitalized with diabetes in Poland. Regular monitoring of the epidemiological data on diabetes-related hospitalization is crucial to assess the effectiveness of diabetes-related care. Moreover, even in the EU, there are multiple different health systems. Data from Poland – the CEE leader – may be interesting for other countries and their health systems specialists.
COMMENT #2: Although it is likely that during Covid-19 pandemic the hospitalization for diabetes related complications (both acute and chronic) reduced, the data presented in this manuscript extremely implausible. In fact, since the poland population consists of 38 milions of individuals, and the polish prevalence of diabetes is about 7%, this results in a approximate number of 2.700.000 diabetes individuals. 95% of these are Type 2 diabetic patients (2.565.000), the rest with T1D, autoimmune disease, (about 135.000 individuals). Even considering the rate of unknown diabetes, that is clearly higher for T2D while highly unlikely for T1D, the data reported by authors are not credible.
RESPONSE #2: Thank you for this comment. We would like to point out the fact that we used an official nationwide registry. This epidemiological analysis is based on data from the Polish National Hospital Register carried out by the National Institute of Public Health – National Institute of Hygiene (NIPH—NIH) within the population-based hospital morbidity study. Discharge reports are submitted to NIPH—NIH by the public and private hospitals (except the psychiatric units) from all administrative regions in Poland. Data are reported using the template specified by Polish law. All the data are verified by the NIPH-NIH experts. The same data source was used in numerous papers previously published in peer-reviewed journals (e.g. https://pubmed.ncbi.nlm.nih.gov/35062280/ https://pubmed.ncbi.nlm.nih.gov/34452324/ https://pubmed.ncbi.nlm.nih.gov/33876908/ https://pubmed.ncbi.nlm.nih.gov/35322962/ https://pubmed.ncbi.nlm.nih.gov/34184506/).
Please note that similar results were obtained by the scientists who analyzed data from the previous years <2014 (https://pubmed.ncbi.nlm.nih.gov/31134776/; https://pubmed.ncbi.nlm.nih.gov/35322962/).
Differences between T2D and T1D. Please note that we discussed this issue in our manuscript – e.g., in line 253 “The hospitalization rate due to type 1 diabetes in Poland is higher than in other EU countries due to the organization of the healthcare in Poland, which obligates physicians to hospitalize new type 1 diabetes patients”.
Nevertheless, we can not exclude some coding/reporting bias. The following statement was added in the limitations section: “A relatively high number of older patients (80 years and over) hospitalized with type 1 diabetes may suggest some coding/diagnosis bias. Nevertheless, we used a nationwide representative registry led by the designated public governmental institution. Further actions are needed to regularly monitor and improve the data quality available in the public medical registries.”
COMMENT #3: In fact, how is possible that a disease almost 100 times more common than another has more or less the same hospitalization rate? Moreover, acute and chronic Type 2 diabetes complications such as glycemic unbalanced, hyperosmolar coma as well as infections, and all the other diabetes related vascular complications account for the most common causes of hospitalizations. Therefore, I think that the data acquired from the authors have a big coding problem.
RESPONSE #3: Please note that we discussed this issue in our manuscript – e.g., in line 253 “The hospitalization rate due to type 1 diabetes in Poland is higher than in other EU countries due to the organization of the healthcare in Poland, which obligates physicians to hospitalize new type 1 diabetes patients”.
Nevertheless, we cannot exclude some coding/reporting bias. The following statement was added in the limitations section: “A relatively high number of older patients (80 years and over) hospitalized with type 1 diabetes may suggest some coding/diagnosis bias. Nevertheless, we used a nationwide representative registry led by the designated public governmental institution. Further actions are needed to regularly monitor and improve the data quality available in the public medical registries.”
However, national registries are widely used in epidemiological research. Please note that this is the first study to assess nationwide data on diabetes-related hospitalizations before and during the pandemic.
COMMENT #4: By the way, this issue is confirmed looking at Table 2 where is stated that between 2014 and 2019 more than 2000 Type 1 diabetes patients aged 80 and older were hospitalyzed every year. I would remind that life expectancy of individuals with type 1 diabetes is about 8-10 years less than general population (Cardiovasc Endoc Metab 2020. Doi: 10.1097/XCE.0000000000000210), therefore to came across in over 80 individuals with T1D is really uncommon, to record thousands of very old people with T1D is very unlikely.
RESPONSE #4: Thank you for this comment. We fully agree with the reviewer that multiple factors may affect the observed phenomenon. Nevertheless, we used official public health registry data. The number of patients hospitalized with T1D and aged 80 is relatively low and does not change over time. As this phenomenon requires further investigation, the limitations section was revised. Moreover, please note that in this study a wide range of analyses was presented that provide practical data for both healthcare professionals as well as policymakers, as stated in the Discussion section. Nevertheless, we will do our best to encourage the management of the registry to improve the quality check procedures.
COMMENT #5: Finally, E12 stands for malnutrition-related diabetes mellitus that is not a type of diabetes, therefore it should be removed from Table 1
RESPONSE #5: Thank you for this comment. As suggested by the Reviewer, ICD-10 code E12 was removed from the Table 1 as well as other parts of the manuscript.
Reviewer 2 Report
The authors presented a study on the trends in diabetes mellitus-related in Poland 2014 and 2020 and to assess changes in these admissions before and during the COVID-19 pandemic using national records. In my opinion, the authors have been comprehensive and meticulous in their writing and analysis. While the topic may not be particularly novel, the findings shed light on the impacts of priorities in the Polish healthcare system. However, I have some concerns regarding the presentation and visualization of results that have made it difficult for the reader to evaluate these findings in meaningful manner. My detailed comments follow below.
Comment 1: [General comment] The COVID-19 pandemic only begun in late 2019 and is still with us today. The pandemic has forced various healthcare systems to shift resources and re-evaluate priorities, but these are fluid and continuously being refined. Thus, would it be worthwhile to include 2021 data?
Comment 2: [Abstract, general comment] Why was only information from 2019 to 2020 included in the Abstract? What about 2014 to 2018?
Comment 3: [Abstract, general comment] Can the authors include P-values whenever appropriate (e.g., from chi-squared tests after converting the percentages to counts)?
Comment 4: [Introduction, general comment] In general the Introduction is meticulously written. However, if word count is an issue, I would suggest the authors be more succinct and focus on areas that the paper directly addresses. For instance, lines 47 to 52 or lines 63 to 70 are not directly relevant to the paper.
Comment 5: [Introduction, line 33] As this is a formal setting, can the authors use “do not” instead of “don't”?
Comment 6: [Results and Tables, general comment] The presentation and visualization of results is a major concern. The tables are difficult to read because of the large number of sheets and numbers. Can the authors consider present these numbers graphically instead? It would also be useful to focus only on key findings, while allotting the full set of the results as Supplemental data. Finally, please also provide appropriate P-values so it is easier for the reader to sift out the main findings of the paper.
Author Response
Thank you very much for all the valuable comments that allow the Authors to increase the scientific soundness of our paper.
COMMENT #1: The authors presented a study on the trends in diabetes mellitus-related in Poland 2014 and 2020 and to assess changes in these admissions before and during the COVID-19 pandemic using national records. In my opinion, the authors have been comprehensive and meticulous in their writing and analysis. While the topic may not be particularly novel, the findings shed light on the impacts of priorities in the Polish healthcare system. However, I have some concerns regarding the presentation and visualization of results that have made it difficult for the reader to evaluate these findings in meaningful manner. My detailed comments follow below.
[General comment] The COVID-19 pandemic only begun in late 2019 and is still with us today. The pandemic has forced various healthcare systems to shift resources and re-evaluate priorities, but these are fluid and continuously being refined. Thus, would it be worthwhile to include 2021 data?
RESPONSE #1: Thank you for this comment. We fully agree with the reviewer that data from the 2021 may be interesting. However, data collection process last 9-12 months. All data submitted by the hospitals to the National Institute of Public Health are verified. Due to this fact, 2021 data will be completed at the end of 2022. Due to this fact, we included only one pandemic year. Nevertheless, in the future we will keep in mind this idea and provide larger retrospective analysis e.g., 5 years after the pandemic onset.
COMMENT #2: [Abstract, general comment] Why was only information from 2019 to 2020 included in the Abstract? What about 2014 to 2018?
RESPONSE #2: Thank you for this comment. Due to the word limit (200 words) we limited abstract to most important finings. However, as suggested by the reviewer the Abstract was revised to provide more data on previous years: “The number of diabetes-related hospitalizations varied from 76220 in 2016 to 45159 in 2020“.
COMMENT #3: [Abstract, general comment] Can the authors include P-values whenever appropriate (e.g., from chi-squared tests after converting the percentages to counts)?
RESPONSE #3: Thank you for this comment. We agree that chi-square test may be used to compare differences in in-hospital mortality rate. However, conversion from percentages to counts may lead to some risk of bias. Due to this fact this conversion was missed.
COMMENT #4: [Introduction, general comment] In general the Introduction is meticulously written. However, if word count is an issue, I would suggest the authors be more succinct and focus on areas that the paper directly addresses. For instance, lines 47 to 52 or lines 63 to 70 are not directly relevant to the paper.
RESPONSE #4: As suggested by the Reviewer, the Introduction section was revised, to focus on the most important background data.
COMMENT #5: Comment 5: [Introduction, line 33] As this is a formal setting, can the authors use “do not” instead of “don't”?
RESPONSE #5: Thank you for this comment. Line 33 was revised. Moreover, the text was checked to replace “don’t” into “do not”.
COMMENT #6: Comment 6: [Results and Tables, general comment] The presentation and visualization of results is a major concern. The tables are difficult to read because of the large number of sheets and numbers. Can the authors consider present these numbers graphically instead? It would also be useful to focus only on key findings, while allotting the full set of the results as Supplemental data. Finally, please also provide appropriate P-values so it is easier for the reader to sift out the main findings of the paper.
RESPONSE #6: Thank you for this comment. As suggested by the Reviewer, a set of figures was provided. Moreover, crude data on numbers of hospitalizations were moved to the supplementary material to improve the visibility of our manuscript.
Reviewer 3 Report
In this study, the authors presented an epidemiological analysis of hospitalization related to diabetes mellitus in Poland between 2014 and 2020, with particular emphasis on (1) trends in diabetes-related hospital admissions; (2) duration of hospitalization; and (3) in-hospital mortality.
The study is interesting because they shows changes in incidence and mortality from diabetes before and during the COVID-19 pandemic. However, the presentation of results is very general, more analysis is required to support the conclusions of their study. I have several major comments.
Statistical analysis
In addition to the crude incidence and mortality, the authors should also estimate the incidence and mortality standardized by age.
Incidence and mortality should be estimated with 95% confidence intervals.
Results
Percentage difference 2019-2020, should be estimated on 2014-2019 projections (expected value in 2020) vs. 2020 (observed value). The authors could use linear regression models to estimate the expected value (Tables 1, 2 and 3).
Table 1, 2 and 3. There are many results, I suggest changing to a figure.
Incidence and mortality should be estimated with 95% confidence intervals (Tables 1, 2 and 3).
Tables 6 and 7. Mortality should be estimated with 95% confidence intervals (Tables 1, 2 and 3). I also suggest estimating excess mortality from type 1 and 2 diabetes. Mortality projections for 2014-2019 (expected value in 2020) should be estimated and compared with 2020 mortality (observed value). Use p-score or 95% CI.
I suggest changing the tables for figures.
Author Response
Thank you very much for all the valuable comments that allow the Authors to increase the scientific soundness of our paper.
COMMENT #1: In this study, the authors presented an epidemiological analysis of hospitalization related to diabetes mellitus in Poland between 2014 and 2020, with particular emphasis on (1) trends in diabetes-related hospital admissions; (2) duration of hospitalization; and (3) in-hospital mortality.
The study is interesting because they shows changes in incidence and mortality from diabetes before and during the COVID-19 pandemic. However, the presentation of results is very general, more analysis is required to support the conclusions of their study. I have several major comments.
RESPONSE #1: Thank you for this comment. The manuscript was revised to increase the scientific soundness of the paper. Please note that we used a nationwide data on diabetes-related hospitalizations over the 7 years.
COMMENT #2: Statistical analysis. In addition to the crude incidence and mortality, the authors should also estimate the incidence and mortality standardized by age.
RESPONSE #2: Thank you for this comment. We fully agree with the reviewer, that there are different methods of data presentation. In this manuscript, we used the same methods as presented in previous papers by Gajewska et al. (Analyses of hospitalization of diabetes mellitus patients in Poland by gender, age and place of residence - PubMed (nih.gov)) and Goryński et al. (Analysis of diabetic patients hospitalizations in Poland by gender, age and place of residence - PubMed (nih.gov)). To maintain data comparability, we decided to use the hospitalization rate per 100,000 inhabitants by age group (lines 121-122). We fully agree that age-standardized rates may be another way of data presentation, however, during these 7 years, there were no markable changes in the structure of the Polish population. Moreover, as noted by the Reviewers, the scope of analysis presented in this study is very vide, so the doubled number of rates presented in this study will complicate the user experience. Moreover, our major aim was to present a trend in diabetes-related hospitalizations which can be sufficiently presented by the current rations. Nevertheless, we will consider age-standardized rates in our further manuscript where a wider range of years will be analyzed (10-years of 20-years analysis). Thank you very much for all the valuable comments that allow the Authors to increase the scientific soundness of our paper.
COMMENT #3: Incidence and mortality should be estimated with 95% confidence intervals.
RESPONSE #3: Thank you for this comment. We fully agree with the reviewer, that there are different methods of data presentation. In this manuscript, we used the same methods as presented in previous papers by Gajewska et al. and Goryński et al. (Analysis of diabetic patients hospitalizations in Poland by gender, age and place of residence - PubMed (nih.gov)). To maintain data comparability, we decided to use the hospitalization rate per 100,000 inhabitants by age group (lines 121-122). After the team discussion on statistical analysis, we believe that the current way of data presentation meets the goals of the study. 95% confidence intervals may be useful, however, in this case of registry-based study, additional columns with 95%CI may have a limited additional value to the scope of the study.
COMMENT #4: Results. Percentage difference 2019-2020, should be estimated on 2014-2019 projections (expected value in 2020) vs. 2020 (observed value). The authors could use linear regression models to estimate the expected value (Tables 1, 2 and 3).
RESPONSE #4: We fully agree with the reviewer, that there are different methods of data presentation. Our major goal was to provide epidemiological analysis that is based on real-life data. We are aware that estimates of the expected values are possible, however, the goal of this study was to provide real-life data. Nevertheless, we will consider estimates and projections as a possible study goal in another scientific project. Moreover, due to the numerous factors that may affect the projection, we would like to avoid overwhelming findings/conclusions, so expected values were missed in this study.
COMMENT #5: Table 1, 2 and 3. There are many results, I suggest changing to a figure.
RESPONSE #5: Thank you for this comment. As suggested by the Reviewer, a set of figures was provided. Moreover, crude data on numbers of hospitalizations were moved to the supplementary material to improve the visibility of our manuscript.
COMMENT #6: Incidence and mortality should be estimated with 95% confidence intervals (Tables 1, 2 and 3).
RESPONSE #6: Thank you for this comment. We fully agree with the reviewer, that there are different methods of data presentation. In this manuscript, we used the same methods as presented in previous papers by Gajewska et al. and Goryński et al. (Analysis of diabetic patients hospitalizations in Poland by gender, age and place of residence - PubMed (nih.gov)). To maintain data comparability, we decided to use the hospitalization rate per 100,000 inhabitants by age group (lines 121-122). After the team discussion on statistical analysis, we believe that the current way of data presentation meets the goals of the study. 95% confidence intervals may be useful, however, in this case of registry-based study, additional columns with 95%CI may have a limited additional value to the scope of the study.
COMMENT #7: Tables 6 and 7. Mortality should be estimated with 95% confidence intervals (Tables 1, 2 and 3). I also suggest estimating excess mortality from type 1 and 2 diabetes. Mortality projections for 2014-2019 (expected value in 2020) should be estimated and compared with 2020 mortality (observed value). Use p-score or 95% CI.
RESPONSE #7: Thank you for this comment. We fully agree with the reviewer, that there are different methods of data presentation. Our major goal was to provide epidemiological analysis that is based on real-life data. We are aware that estimates of the expected values are possible, however, the goal of this study was to provide real-life data. Nevertheless, we will consider estimates and projections as a possible study goal in another scientific project. Moreover, due to the numerous factors that may affect the projection, we would like to avoid overwhelming findings/conclusions, so expected values were missed in this study.
COMMENT #8: I suggest changing the tables for figures.
RESPONSE #8: As suggested by the Reviewer, a set of figures was provided. Moreover, crude data on numbers of hospitalizations were moved to the supplementary material to improve the visibility of our manuscript.
Round 2
Reviewer 2 Report
I think the presentation of the results has substantially improved with the addition of the graphs. However, without P-values (e.g., from chi-squared tests) or confidence intervals (e.g., from regression analyses), it may be difficult to judge if the differences have been statistically significant.
Author Response
In response to the comments from Reviewer #2
Thank you very much for all the valuable comments that allow the Authors to increase the scientific soundness of our paper.
COMMENT #1: I think the presentation of the results has substantially improved with the addition of the graphs. However, without P-values (e.g., from chi-squared tests) or confidence intervals (e.g., from regression analyses), it may be difficult to judge if the differences have been statistically significant.
RESPONSE #1: Thank you for this comment. As suggested by the Reviewer, Chi-square tests were applied in all applicable places. Tables 1, 4 and 5 were revised and new columns with p-value were added. All changes are marked green.
Reviewer 3 Report
No comments
Author Response
In response to the comments from Reviewer #3
COMMENT #1: No comments
RESPONSE #1: Thank you for reviewing our manuscript and all the comments that allowed us to improve the scientific soundness of the manuscript.